# A Bioinformatics Investigation of Hub Genes Involved in Treg Migration and Its Synergistic Effects, Using Immune Checkpoint Inhibitors for Immunotherapies

**DOI:** 10.3390/ijms25179341

**Published:** 2024-08-28

**Authors:** Nari Kim, Seoungwon Na, Junhee Pyo, Jisung Jang, Soo-Min Lee, Kyungwon Kim

**Affiliations:** 1Biomedical Research Center, Asan Institute for Life Sciences, Asan Medical Center, Seoul 05505, Republic of Korea; nari.kim.0908@gmail.com (N.K.); 87nasw@gmail.com (S.N.); 2College of Pharmacy, Chungbuk National University, Cheongju 28644, Republic of Korea; stdpjh@naver.com; 3Trial Informatics Inc., Seoul 05544, Republic of Korea; etmira8787@gmail.com; 4Samjin Pharmaceutical Co., Ltd., Seoul 04054, Republic of Korea; soominlee@samjinpharm.co.kr; 5Departments of Radiology and Research Institute of Radiology, Asan Medical Center, College of Medicine, University of Ulsan, Olymphic-ro 43 Gil 88, Songpa-gu, Seoul 05505, Republic of Korea

**Keywords:** regulatory T cell, Treg migration, immune response

## Abstract

This study aimed to identify hub genes involved in regulatory T cell (Treg) function and migration, offering insights into potential therapeutic targets for cancer immunotherapy. We performed a comprehensive bioinformatics analysis using three gene expression microarray datasets from the GEO database. Differentially expressed genes (DEGs) were identified to pathway enrichment analysis to explore their functional roles and potential pathways. A protein-protein interaction network was constructed to identify hub genes critical for Treg activity. We further evaluated the co-expression of these hub genes with immune checkpoint proteins (PD-1, PD-L1, CTLA4) and assessed their prognostic significance. Through this comprehensive analysis, we identified CCR8 as a key player in Treg migration and explored its potential synergistic effects with ICIs. Our findings suggest that CCR8-targeted therapies could enhance cancer immunotherapy outcomes, with breast invasive carcinoma (BRCA) emerging as a promising indication for combination therapy. This study highlights the potential of CCR8 as a biomarker and therapeutic target, contributing to the development of targeted cancer treatment strategies.

## 1. Introduction

Regulatory T cells (Tregs) play an important role in maintaining immune homeostasis and preventing autoimmunity by suppressing excessive immune responses [1]. Tregs are most characterized by the expression of CD3+CD4+CD25+FoxP3+ markers in both mice and humans [2]. Tregs are composed of several subsets, primarily classified into two main categories: naturally occurring Tregs (nTregs) and peripheral Tregs (pTregs). nTregs develop in the thymus and are characterized by markers such as CD4+CD25+Foxp3+Helios+CTLA4+Nrp1+. In contrast, pTregs are induced in the periphery, sharing similar markers with nTregs, but differing in their origin [3]. FOXP3 has long been regarded as the ‘master regulator’ of Treg cells, controlling their differentiation and immunosuppressive functions [4]. Elkord et al. demonstrated the crucial role of FOXP3 as a transcription factor in CCR8+ Treg cells by showing that FOXP3 up-regulation enhances the suppressive functions of these cells in both in vitro human cell assays and in vivo models of autoimmune disease [5].

In the tumor microenvironment (TME), Tregs promote immune suppression, thereby preventing effector T cells from effectively targeting and killing cancer cells [6,7,8]. In this space, Treg cells migrate to sites of inflammation and suppress various types of effector lymphocytes, including CD4+ helper T cells and CD8+ cytotoxic T lymphocytes [9]. Therefore, targeting Treg cells may be a promising strategy because it can reduce immunosuppression, activate the immune environment, and improve the effectiveness of other treatments [10].

Several therapeutic strategies have been developed and studied to enhance antitumor immunity, including depleting Treg cells or blocking their migration to lymph nodes or tumors [11]. One notable example is mogamulizumab, also called Poteligeo, a monoclonal antibody targeting CCR4 [12]. CCR4 is highly expressed on tumor-infiltrating Tregs, contributing to their migration and accumulation in the tumor microenvironment [13]. Mogamulizumab is used in the treatment of cutaneous T-cell lymphomas (CTCLs) and works by binding to CCR4, reducing tumor burden by inhibiting Treg migration and survival [14]. By binding to the receptor, mogamulizumab induces antibody-dependent cellular cytotoxicity (ADCC), leading to the depletion of Tregs and the enhancement of anti-tumor immune responses [15]. This drug was evaluated based on a randomized, open-label, multicenter trial (NCT no.; NCT01728805) in patients with active mycosis fungoides (MF) or Sézary syndrome (SS), two rare forms of non-Hodgkin lymphoma, after at least one prior systemic therapy [16]. The phase III MAVORIC trial demonstrated that mogamulizumab significantly improved progression-free survival (PFS) and overall response rate (ORR) compared to the comparator, vorinostat [12]. Specifically, the median PFS was 7.7 months for mogamulizumab versus 3.1 months for vorinostat, while the ORR was significantly higher for mogamulizumab at 28% compared to 5% for vorinostat [15,17]. Another promising approach is to enhance the immune response to tumors by inhibiting chemokine–chemokine-receptor interactions, thereby preventing Tregs from accumulating around cancer cells [18,19]. Recent studies have shown that the CXCR4 antagonist AMD3100, also known as plerixafor (Marketed Mozobil^®^ (Sanofi, Paris, France)), effectively blocks the CXCL12/CXCR4 interaction, thereby inhibiting the migration of Tregs and other immune cells within the tumor microenvironment [20]. The primary ligand of CXCR4 is CXCL12, also known as SDF-1 (Stromal Cell-Derived Factor-1). CXCL12 attracts cancer cells by binding to CXCR4, which is expressed on both hematopoietic and non-hematopoietic tumor cells [21]. This interaction not only facilitates tumor progression by promoting cancer cell migration to favorable niches, but also contributes to immune suppression within the TME [22]. Mozobil^®^ was approved by the FDA for this indication in patients with non-Hodgkin’s lymphoma or multiple myeloma [23]. This mechanism not only enhances the efficacy of immunotherapy by preventing tumor immune evasion, but also demonstrates significant potential in mobilizing hematopoietic stem cells for transplantation [24]. One consideration is that when designing studies targeting Tregs it is essential to consider that their role may differ between solid tumors and hematological malignancies, particularly lymphoid neoplasms, where their functional roles may vary considerably [25]. In solid tumors, Tregs primarily promote immune suppression and facilitate tumor growth [26], whereas in lymphoid neoplasms their role can vary from contributing to disease progression to maintaining immune homeostasis [27]. 

The infiltration of immune cells into solid tumors, their migration within the TME, and interactions with other immune cells are regulated by chemokine gradients [28]. The chemokine family consists of small, secreted proteins structurally similar to cytokines, and is divided into four families (CCL, CXCL, CL, and CX3CL) which regulate immune cell migration [29]. Inhibiting the interaction between chemokines and chemokine receptors can reduce the infiltration of Tregs into tumor sites, thereby reactivating suppressed anti-tumor immune reactions [30]. Chemokine receptors are G protein-coupled receptors (GPCRs) which are subdivided into four subfamilies: CXCR, CX3CR, XCR, and CCR, based on the class of chemokines they bind [31]. When chemokines bind to their respective receptors, they can promote tumor growth by promoting the recruitment and maintenance of immunosuppressive cells such as Tregs [32]. Strategies that target Tregs utilizing the chemokine system may be highly valuable for immunotherapy, given its role in leukocyte recruitment, activation, angiogenesis, cancer cell proliferation, and metastasis [33]. 

However, these approaches have limitations, including the potential for off-target effects and the complexity of the chemokine network involved in Treg migration [34]. Firstly, the lack of specificity in targeting Tregs is a significant challenge, as Tregs not only function in the tumor microenvironment but also in maintaining immune homeostasis in normal tissues [35]. Furthermore, selecting accurate targets for Tregs is difficult because markers such as FOXP3, CTLA-4, and PD-1 overlap with those of other immune cells [36]. Secondly, the complexity of the immune system, involving the interaction of multiple immune cell types and a variety of molecules, introduces significant limitations for targeting Tregs [37]. Modulating Treg activity could unexpectedly impact other immune components, potentially disrupting the tumor microenvironment and overall immune balance [38]. Therefore, it is important to utilize bioinformatics techniques to understand the functions of Tregs and related hub genes in the drug development stage or preclinical trial stage [39,40]. This approach can improve the accuracy and safety of Treg-targeting strategies and minimize unexpected immune responses.

Hub gene identification refers to screening key genes that form the central part of gene networks due to their interactions with other genes [40]. This is important because hub genes play a fundamental role in biological processes and disease mechanisms [41]. Recent studies have highlighted the importance of hub gene identification in cancer research [42,43]. For instance, a similar study identified eight key hub genes (CDK1, KIF11, CCNA2, TOP2A, ASPM, AURKB, CCNB2, and CENPE) in breast cancer, linking them to pathways such as focal adhesion and ECM–receptor interaction [44]. Their survival analysis further demonstrated the potential of these genes as prognostic and diagnostic biomarkers, underscoring their relevance in improving breast cancer prognosis and treatment strategies [42].

In this study, we obtained and analyzed three original microarray gene expression datasets, GSE128822, GSE116347, and GSE120280, from the GEO database, to investigate the function of Tregs. The flowchart for this study procedure is shown in Figure 1. Differentially expressed genes (DEGs) were identified, followed by a series of bioinformatics approaches to identify hub genes and perform enrichment analysis. The identification of these common DEGs across different datasets underscores potential shared molecular pathways and regulatory mechanisms that could be crucial in understanding Treg functionality in various cancer types [45]. Additionally, we analyzed the potential synergies between these key genes and immune checkpoint proteins from an immunotherapy perspective using public databases. This comprehensive bioinformatic analysis provides the foundation for further investigations into the specific roles of these genes and their contributions to Treg-mediated immune responses.

## 2. Results

### 2.1. GEO Dataset Selection

This study involved downloading three publicly available gene sets (GSE128822, GSE116347, and GSE120280) from the GEO database, specifically focused on Treg-related gene expression profiles. The GEO platform includes GPL18573 (Illumina NextSeq 500 (Illumina Inc., San Diego, CA, USA), *Homo sapiens*) and GPL19057 (Illumina NextSeq 500, Mus musculus). The GSE128822 dataset involves gene expression profiles of FACS-sorted Tregs (CD4+CD25+CD127-CCR8+ICOS+ and CD4+CD25+CD127-CCR8-ICOS-) from five surgically resected and cryopreserved human non-small-cell lung cancer carcinomas. The GSE116347 dataset consists of CD4+ Treg cells from 12 surgically resected and cryopreserved human colorectal carcinomas or normal colon tissues, classified into Tumor Tregs and Normal Tregs. The GSE120280 dataset includes CD4+GFP+ Tregs and CD4+GFP- Resting conventional T cell (Tconv) populations, with spleen and tumor samples harvested and processed into single cells from tumor-bearing mice and the spleens of normal mice (Table 1).

### 2.2. Identification of DEGs

Overall, 57 differentially expressed genes (DEGs) were identified (35 upregulated genes and 22 downregulated genes). As shown in the heatmap (Figure 2a), the patterns of common gene expression based on log_2_FoldChange values were consistent between the Treg group and the non-Treg group across the three datasets. The Venn diagram (Figure 2b) illustrates the overlap of DEGs among the three datasets, identifying common genes shared across all datasets and unique genes specific to each dataset. The volcano plot visualizing these DEGs, including their significance and fold change, is shown in Figure 3. The list of overlapping genes identified in the differential expression analysis, including their classification as upregulated or downregulated and associated expression metrics, is available in Appendix A.

### 2.3. GO, KEGG and Reactome Pathway Enrichment Analysis

Gene Ontology (GO) analysis revealed that the differentially expressed genes (DEGs) are predominantly involved in immune-related functional categories In the GO Functional Analysis figure, the most prominent observation is the significant enrichment of biological process (BP) categories, as indicated by the large number of red bars. GO analysis revealed that upregulated genes were significantly enriched in immune-related processes such as cytokine receptor activity, T cell activation, and cytokine production, as well as being associated with the external side of the plasma membrane (Figure 4a). Downregulated genes (Figure 4b) were associated with processes like platelet activation and neutrophil chemotaxis, although these pathways are less directly relevant to Treg function. 

The KEGG pathway enrichment analysis further supports the involvement of DEGs in immune regulation. Figure 5 illustrates the KEGG pathway enrichment results for upregulated genes, highlighting significant pathways such as ‘Cytokine-mediated signaling pathway’ (Figure 5a) and ‘Cytokine–cytokine-receptor interaction’ (Figure 5b). No KEGG pathways were significantly enriched for downregulated genes. 

Our Reactome Pathway Enrichment Analysis identified several key pathways significantly enriched among the DEGs associated with Treg migration (Table 2). Notably, the ‘RUNX1 and FOXP3 control the development of regulatory T lymphocytes (Tregs)’ pathway (R-HSA-8877330, *p*-value: 6.27 × 10^−4^) underscores the crucial roles of these transcription factors in Treg function and migration. 

From the enrichment analysis of DEGs identified in selected GEO datasets, we provide a comprehensive understanding of the molecular mechanisms that regulate immune responses, with significant involvement in pathways important for immune regulation and Treg function.

### 2.4. Hub Gene Identification from the PPI Network

In our study, we constructed a protein–protein interaction (PPI) network to identify hub genes involved in Treg function (Table 3, Figure 6). The analysis highlighted IL2RA, TNFRSF4, TNFRSF18, and CCR8 as central players in this network. IL2RA exhibited the highest degree and betweenness centrality, underscoring its critical role in Treg cell function. 

The analysis revealed several key genes based on degree centrality, betweenness centrality, closeness centrality, and clustering coefficient (Table 3). This presents the centrality measures for key genes identified in the PPI network related to Treg function. 

The hub genes identified include IL2RA, TRAF1, IL1R2, BATF, IL12RB2, CD80, TNFRSF4, TNFRSF18, TRAF3, and CCR8. Specially, IL2RA showed the highest values in degree (11), betweenness (0.29), and closeness centrality (0.76), indicating its central role in the PPI network. CCR8, with its high clustering coefficient (0.81), suggests significant involvement in a specific module, emphasizing its potential importance in Treg function and protein interactions. These genes were categorized based on their roles in immune regulation, cytokine signaling, chemokine signaling, and transcriptional regulation. Table 4 presents a detailed classification of these hub genes, highlighting their specific functions and contributions to Treg cell behavior and immune response modulation.

Our analysis of GEO datasets has identified key hub genes involved in the diverse functional aspects of Treg cells, including immune regulation, cytokine and chemokine signaling, Treg cell migration, and transcription factor activities. Notably, CCR8 emerged as a key player in Treg migration within the tumor microenvironment. The high clustering coefficient of CCR8 in the PPI network underscores its significant role in chemokine-mediated processes.

### 2.5. Synergistic Effects of CCR8 and Immune Checkpoint Inhibitors

#### 2.5.1. Correlation Analysis between CCR8 Expression and Immune Checkpoint Proteins

The scatter plots in Figure 7 demonstrate the correlation between CCR8 expression levels and the expression levels of several key immune genes (PDCD1 (PD-1), PD-L1 (CD274), CTLA4) across various cancer types. The analysis reveals significant positive correlations, indicating that higher CCR8 expression is associated with increased expression of genes related to immune response. For all three genes, the highest correlations with CCR8 expression were observed in BRCA (Breast Invasive Carcinoma). The correlation coefficients (cor) and *p*-values are as follows: PDCD1 with a cor of 0.584 and *p*-value of 9.67 × 10^−102^ (Figure 7a), CD274 with a cor of 0.606 and *p*-value of 4.12 × 10^−63^ (Figure 7b), and CTLA4 with a cor of 0.758 and *p*-value of 2.64 × 10^−206^ (Figure 7c).

#### 2.5.2. TMB, MSI Correlation Analysis

Figure 8 shows a radar chart showing the correlation between CCR8 expression and TMB and MSI across different cancer types. The analysis shows that CCR8 expression is correlated with TMB and MSI in several cancers, suggesting that higher CCR8 levels may be associated with increased genomic instability and mutation rates in tumors.

In the radar plots analyzing correlations between gene expression and TMB and MSI, CCR8, CD274, and PDCD1 expression levels showed significant positive correlations with TMB and MSI in several cancer types. Notably, colon adenocarcinoma (COAD) and uterine corpus endometrial carcinoma (UCEC) displayed strong correlations across all three genes. Breast invasive carcinoma (BRCA) and stomach adenocarcinoma (STAD) also exhibited notable correlation.

### 2.6. Comprehensive Integration of Multiple Analysis

In our comprehensive analysis, we integrated multiple analytical methods to evaluate the potential effectiveness of CCR8 and immune checkpoint inhibitor (ICI) combination therapy across six selected cancer types (Table 5). 

BRCA is the most promising indication for CCR8 and ICI combination therapy. The high final score of 30 reflects significant correlations with immune checkpoint markers, Treg infiltration, high TMB and MSI, and those included in current standard ICI treatment protocols.

## 3. Discussion

Our comprehensive bioinformatics analysis investigated hub genes associated with Treg migration and explored potential synergies with ICIs. We hypothesized that identifying these hub genes and their networks could lead to the discovery of novel therapeutic targets to enhance anti-tumor immune responses within the complex tumor microenvironment using GEO datasets. We focused on the functions of Tregs, particularly highlighting key genes for Treg migration, and demonstrated their potential synergistic effects with ICIs from an immunotherapy perspective. 

In this study, we identified several hub genes associated with Treg migration and function, emphasizing their critical roles in cytokine signaling and immune regulation within the TME. Key genes such as IL2RA, CCR8, CD80, and TNFRSF18 are involved in Treg proliferation, migration, and immune modulation, highlighting their potential as therapeutic targets. IL2RA and CCR8 are particularly notable for their roles in Treg survival and chemokine-mediated migration, respectively, offering approaches to inhibit Treg-mediated immune suppression. Supporting our findings, previous research has demonstrated that CD4+CD25 (known as IL2RA)+ Tregs play a crucial role in immune regulation [57]. By isolating these cells and conducting flow cytometry and suppression assays, they showed that CD25+ Tregs effectively suppress the proliferation of CD4+CD25− effector T cells. Their work supports our observation that modulating CD25+ Tregs can significantly impact immune homeostasis. CCR8 expression is upregulated in Treg cells, following activation in the presence of CCL1, which enhances their suppressive activity through a STAT3-dependent pathway, including upregulation of FOXP3, CD39, IL-10, and granzyme B [5]. These findings are consistent with our observation that CCR8 plays a critical role in Treg-mediated immune regulation, highlighting the therapeutic potential of targeting the CCL1–CCR8 axis to modulate Treg function and improve cancer immunotherapy outcomes. The identification of these genes highlights the importance of cytokine pathways in Treg function and suggests that targeting these pathways could enhance the efficacy of ICIs [5]. We observed a positive correlation between CCR8 expression and immune checkpoint-protein expression in the TCGA dataset; however, further studies are needed to specifically investigate the co-interactions and correlations between PD-L1/PD-1, CTLA-4, and CCR8+Treg to validate these findings. Although relevant research, such as the development of CCR8/CTLA-4 bispecific antibodies [58] and combination strategies targeting CCR8 and PD-1/PD-L1 [59], is currently underway, a detailed exploration of the direct correlations between these markers remains necessary.

The strength of this study is its data-driven approach, featuring comprehensive bioinformatics analysis, and its ability to suggest therapeutic indications for cancer treatment [60]. This approach offers a cost-effective and scalable method to screen large datasets, predict drug targets, and explore indications for personalized medicine [61]. Furthermore, predictive modeling using machine learning enhances the ability to simulate and predict drug efficacy and therapy responses [62]. However, the study has several limitations, including the lack of experimental validation to confirm the bioinformatics findings [63]. These data sets primarily focus on gene expression data, and may not include clinical characteristics of patients. The lack of detailed clinical and pathological data, such as patient survival and efficacy information, in these datasets limits the applicability of gene expression results to clinical trials. Moreover, the complexity of the TME and tumor heterogeneity presents significant challenges with respect to analysis results being fully representative [64]. The TME contains a variety of cell types with complex interactions, and tumors often display genetic and cellular heterogeneity [65], resulting in differences in behavior and therapeutic response. As a result, bioinformatics analyses may not represent all aspects of this complexity, highlighting the need for complementary experimental validation and personalized approaches [66].

Future research will focus on conducting a comparative analysis of genes specifically regulated within T cell subsets in the TME, comparing these findings with the results of this study, to further validate our conclusions. Additionally, using our computational analysis techniques, we are simulating how modulating Tregs within the TME impacts tumor growth dynamics and overall progression, aiming to enhance our understanding of therapeutic strategies targeting Treg-mediated immune regulation.

## 4. Materials and Methods

### 4.1. GEO Dataset Collection

In this study, we analyzed datasets from the Gene Expression Omnibus (GEO) database (https://www.ncbi.nlm.nih.gov/geo/, accessed on 25 July 2024). We queried the GEO datasets using the keywords “Treg” or “Regulatory T cell”. From the search results, we selected tumor tissue microarray data without restricting them to any specific type of cancer, and focused on datasets. Using the GEOquery library in R, we verified the presence of both expression data and phenotype data before downloading the datasets.

### 4.2. Identification of DEGs in GEO Datasets

To identify differentially expressed genes (DEGs) in our datasets, we utilized the DESeq2 package in R [67]. The analysis was conducted as follows: we loaded the downloaded expression files and metadata files, preprocessing the expression data using the make.unique function to ensure unique gene names. A DESeq2 dataset was then created using the DESeqDataSetFromMatrix function, which required count data, metadata, and a design. The design is set to a value that exists in both the metadata and the expression data, and it is used to create different groups (Treg vs. non-Treg). The DESeq2 analysis was performed by calling the DESeq function on the dataset object, which estimates size factors and dispersions, and conducts negative binomial GLM fitting and Wald significance tests [68]. Finally, the results of the differential expression analysis were extracted using the results function, providing log_2_ fold changes, *p*-values, and adjusted *p*-values for each gene. Subsequently, genes that did not satisfy the criteria of log_2_|fold change (FC)| ≥ 0.5 and *p*-value < 0.05 were filtered out.

### 4.3. GO, KEGG and Reactome Pathway Enrichment Analysis

The Gene Ontology (GO) database (https://geneontology.org, accessed on 25 July 2024) is utilized for gene annotation and the analysis of biological processes [69]. GO terms were categorized into three types: biological process (BP), molecular function (MF), and cellular component (CC). The Kyoto Encyclopedia of Genes and Genomes (KEGG) database integrates information on genomic [70], chemical, and system functions. The “clusterProfiler” [71] package in R was utilized for GO annotation and KEGG pathway enrichment analyses of the overlapping genes. A statistically significant enrichment was indicated by *p*  <  0.05.

The pathway enrichment analysis was conducted using the ReactomePA package in R [72], identifying significantly enriched pathways among the differentially expressed genes (DEGs) involved in immune regulation and Treg function.

### 4.4. Hub Gene Identification from the PPI Network

To obtain protein–protein interaction (PPI) information, the identified DEGs were submitted to the STRING database (STRING database, version 11.5, available at https://string-db.org/, maintained by the Swiss Institute of Bioinformatics (SIB), Lausanne, Switzerland) and imported into Cytoscape (Cytoscape, version 3.8.2, an open-source software platform for visualizing complex networks, available at https://cytoscape.org/, developed by the Cytoscape Consortium), an open-source software platform for visualizing molecular interaction networks. Cytoscape’s NetworkAnalyzer tool [73] was used to calculate key centrality measures for each node in the network, including degree centrality, betweenness centrality, and closeness centrality. Genes with the highest centrality scores were identified as hub genes, focusing particularly on those with high degree and betweenness-centrality scores, due to their likely critical regulatory roles within the network. Among the hub genes identified through the PPI network, we specifically selected genes related to the chemokine signaling pathway involved in Treg migration.

### 4.5. Synergistic Effects of CCR8 and Immune Checkpoint Inhibitors

To explore the potential synergy between hub genes (CCR8) associated with Treg function and the known immune checkpoint proteins PD-L1, PD-1, and CTLA4, we analyzed their expression correlations across six selected cancer types: Breast invasive carcinoma (BRCA), Head and Neck squamous cell carcinoma (HNSC), Colon adenocarcinoma (COAD), Stomach adenocarcinoma (STAD), Thyroid carcinoma (THCA), and Rectum adenocarcinoma (READ). These cancer types were selected based on our previous work, which comprehensively analyzed the potential targeting potential of CCR8 [74]. First, the GEPIA platform (GEPIA, available at http://gepia.cancer-pku.cn/, developed by the Bioinformatics Department of Peking University, Beijing, China) was used to compare the expression levels of immune checkpoint proteins between normal and cancer tissues, using ACC as a control. Second, we used the TIMER database (TIMER, available at https://cistrome.shinyapps.io/timer/, developed by the Cistrome Project, Harvard University, Boston, MA, USA) to assess correlations between Treg migration-related genes and immune checkpoint genes within these cancer types.

Analyzing the correlation between single gene expression and TMB, as well as MSI, is critical, as these are key biomarkers for predicting immunotherapy response and understanding tumor immunogenicity. We performed a pan-cancer analysis of the correlation between CCR8, PDCD1, and CD274 expression and both tumor mutational burden (TMB) and microsatellite instability (MSI) using the TCGAplot R package [75], which facilitates comprehensive multi-omics data analysis from The Cancer Genome Atlas (TCGA). Radar plots were created to display these correlations, illustrating the strength and significance of correlations between gene expression levels and cancer biomarkers across various cancer types, with each axis representing a different cancer and asterisks (*) indicating statistical significance. The abbreviations for the TCGA cancers are available in the Appendix A.

### 4.6. Comprehensive Integration of Multiple Analysis

To systematically evaluate the potential effectiveness of combination therapies targeting CCR8 and ICIs, we conducted an integrated analysis across the six selected cancer types [74]. This analysis integrates correlation data from CCR8 with immune checkpoint genes, Treg infiltration, TMB, and MSI, along with clinical relevance (Table 6). 

Scores from each analysis were summed to calculate a total score for each cancer type. We also considered the clinical status of immune checkpoint inhibitors (ICIs) for these cancer types, assigning 3 additional points for cancers where ICIs are the currently used standard of care (SOC). Additional weight was given to cancer types with high TMB, to reflect their increased responsiveness to immunotherapy.

## 5. Conclusions

In summary, we analyzed three GEO datasets using comprehensive bioinformatics approaches to identify Treg-associated genes involved in immune modulation. Our study identified ten hub genes, including IL2RA, CCR8, TNFRSF4, TNFRSF18, and CD80, which play crucial roles in Treg function and migration. The findings emphasize the significance of the CCR8 as potential therapeutic targets and prognostic markers for enhancing cancer immunotherapy. The correlation of CCR8 with immune checkpoint proteins and genomic instability markers such as TMB and MSI further supports its potential in combination therapies for various cancer types, particularly breast invasive carcinoma (BRCA).

## Figures and Tables

**Figure 1 ijms-25-09341-f001:**
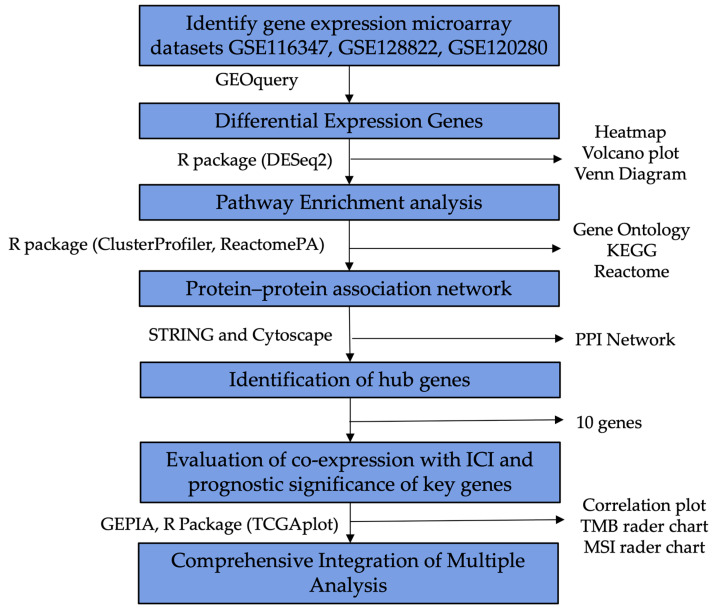
Flowchart of study process.

**Figure 2 ijms-25-09341-f002:**
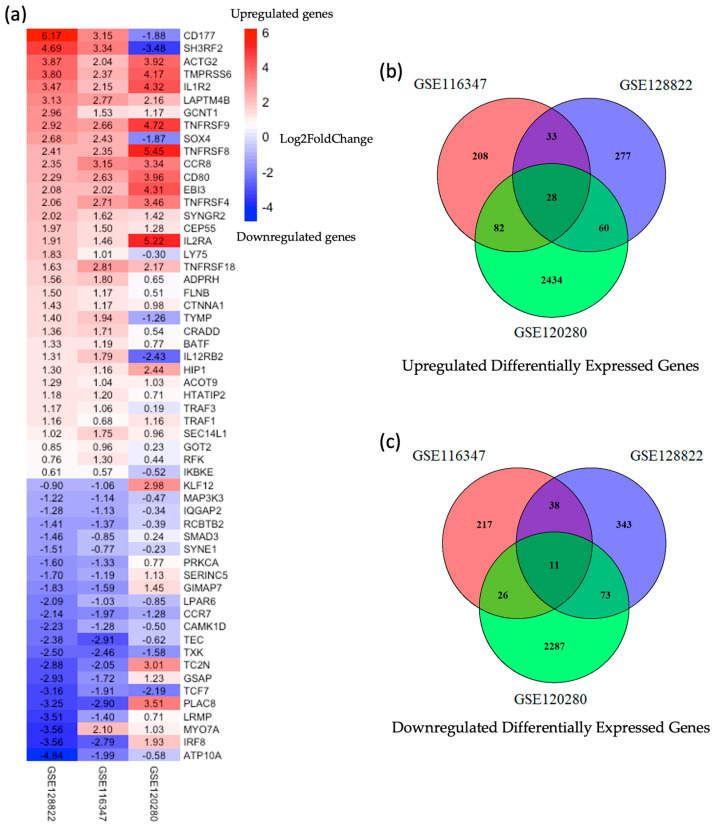
GEO dataset of differentially expressed genes (DEGs). (**a**) The heatmap shows the patterns of common gene expression. The heatmap uses a color scale where red indicates upregulated genes (positive Log_2_FoldChange), and blue indicates downregulated genes (negative Log_2_FoldChange). (**b**,**c**) Venn diagrams illustrate the number of common genes shared by the three GEO datasets. The top Venn diagram shows the intersection of upregulated genes (**b**), while the bottom one shows the intersection of downregulated genes (**c**).

**Figure 3 ijms-25-09341-f003:**
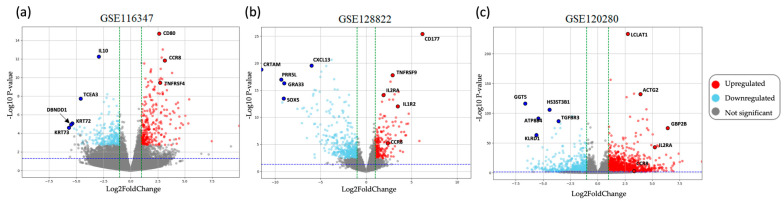
Volcano plot of DEGs in datasets from GEO database. (**a**–**c**) Volcano plot of DEGs in each dataset, highlighting the significance (*p*-value) and magnitude (log_2_FoldChange) of gene expression changes in each respective dataset. In these plots, downregulated genes are represented in light blue, upregulated genes in red, and non-significant genes in gray.

**Figure 4 ijms-25-09341-f004:**
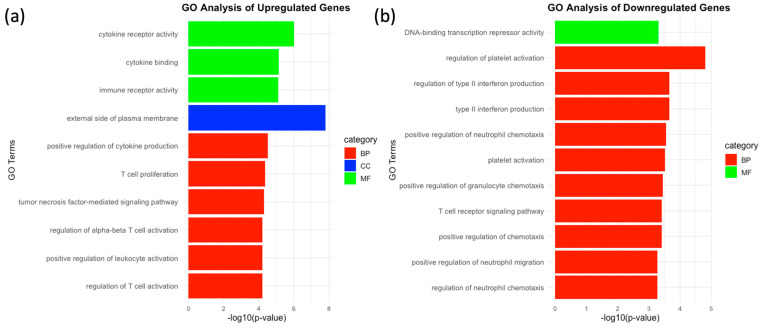
(**a**,**b**) GO Functional Analysis of Differentially Expressed Genes (DEGs). In the BP category, immune-related processes such as cytokine receptor activity, regulation of cell adhesion, and T cell activation were prominently enriched. The CC category primarily included genes associated with the plasma membrane, while the MF category highlighted functions like G protein-coupled receptor activity and cytokine binding.

**Figure 5 ijms-25-09341-f005:**
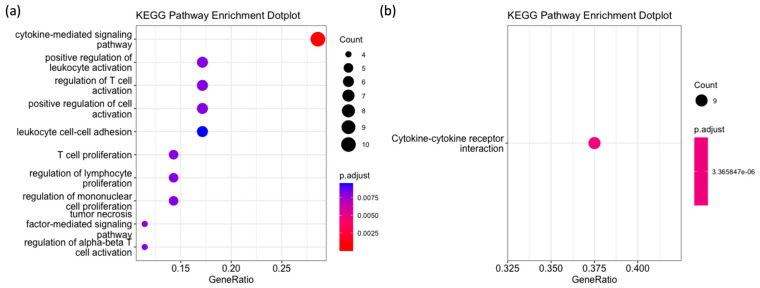
Results of KEGG pathway analysis. (**a**) KEGG pathway enrichment dot plot showing significant pathways with gene ratio, count, and adjusted *p*-values. (**b**) Detailed view of the ‘Cytokine–cytokine-receptor interaction’ pathway, emphasizing its statistical significance.

**Figure 6 ijms-25-09341-f006:**
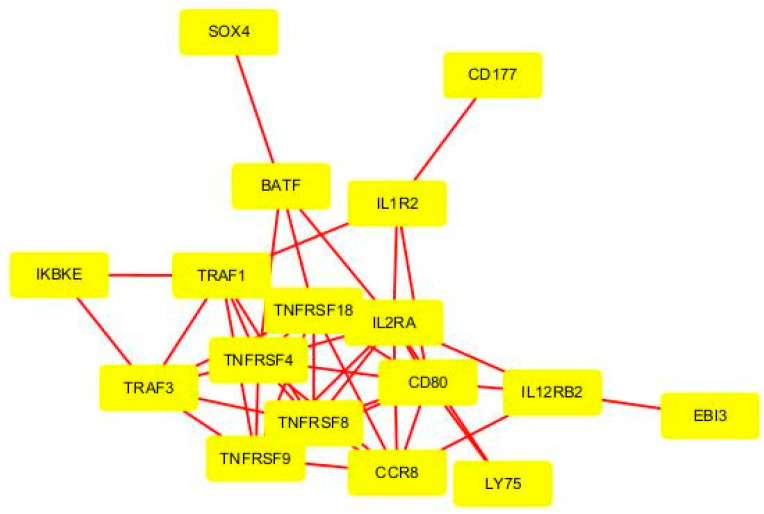
Protein–Protein Interaction (PPI) Network of Key Genes Involved in Treg Function. Yellow nodes represent individual genes, and edges (red lines) represent interactions between these genes.

**Figure 7 ijms-25-09341-f007:**
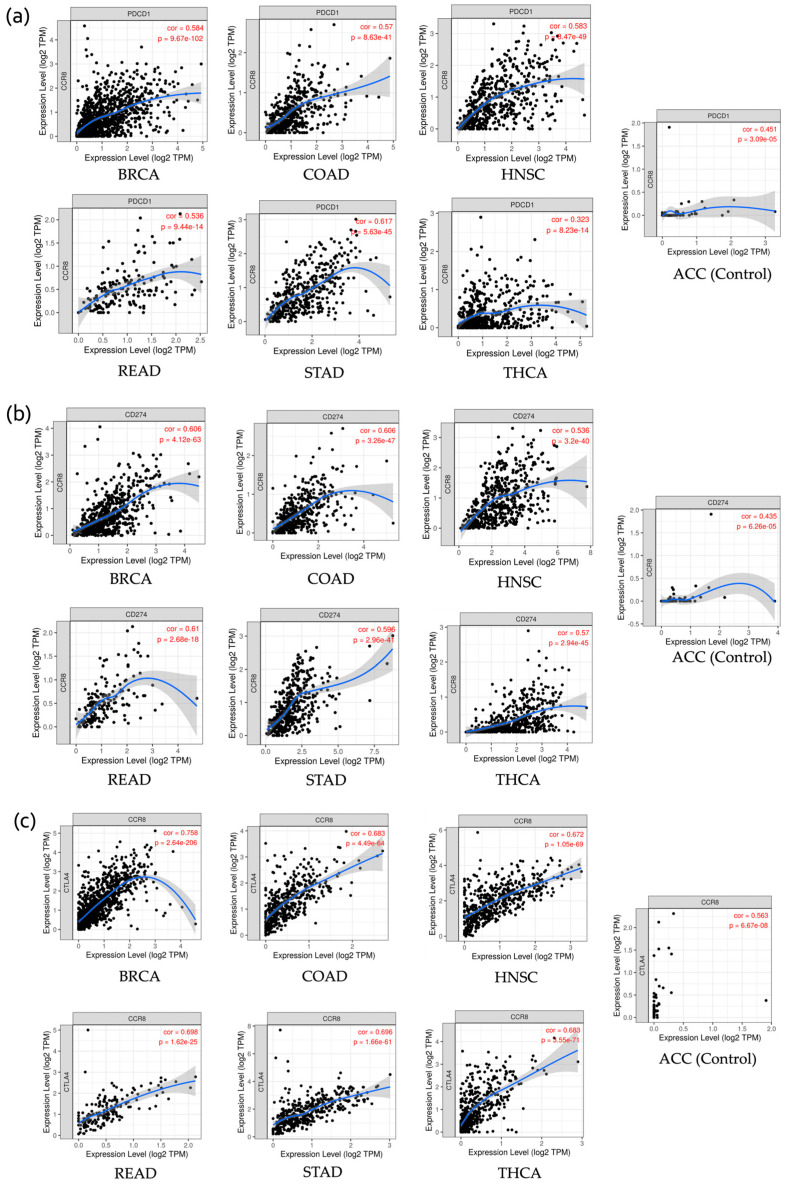
Correlation Analysis of CCR8 Expression with Other Immune Checkpoint Proteins (PDCD1, CD274, CTLA4) Across Various Cancer Types. Each figure shows that CCR8 expression levels are positively correlated with these immune checkpoint proteins, with correlation coefficients and *p*-values indicating statistical significance in the respective cancer types. Figure (**a**) shows the correlation with PDCD1 (PD-1), (**b**) with CD274 (PD-L1), and (**c**) with CTLA4. Abbreviation: cor, correlation coefficient; *p*, *p*-value.

**Figure 8 ijms-25-09341-f008:**
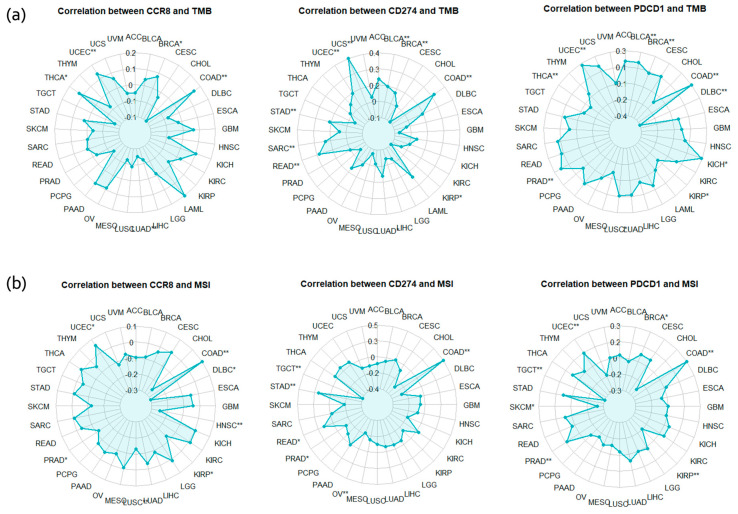
Correlation Analysis of TMB (**a**) and MSI (**b**) with CCR8 and Immune Checkpoint Proteins (CD274, PDCD1) Across Various Cancer Types. In the radar plots, the single asterisk (*) indicates a statistically significant correlation between gene expression and TMB or MSI, with a *p*-value less than 0.05, while the double asterisk (**) denotes a highly significant correlation, with a *p*-value less than 0.01.

**Table 1 ijms-25-09341-t001:** Selected GEO dataset for bioinformatics analysis.

Accession No.	Platform	Total Samples	Case (Treg Group)	Control (Non-Treg Group)	Species	Disease/Condition	Pubmed ID
GSE128822	GPL18573	10	5	5	*Homo sapiens*	Lung adenocarcinoma	32125291, 34552178
GSE116347	GPL18573	18	12	6	*Homo sapiens*	Colorectal carcinoma	30348759
GSE120280	GPL19057	12	6	6	Mus musculus	Lung cancer	32878747

Abbreviations: GEO, gene expression omnibus; GSE, GEO series; GPL, GEO platform; Treg, Regulatory T cell.

**Table 2 ijms-25-09341-t002:** Results of Reactome Pathway Enrichment Analysis. Pathway ID is a unique identifier in the Reactome database, Description provides a brief overview of the pathway, *p*-value shows statistical significance, and Count indicates the number of DEGs involved in each pathway.

Pathway ID	Description	*p*-Value	Count
R-HSA-5669034	TNFs bind their physiological receptors	4.00 × 10^−6^	4
R-HSA-5668541	TNFR2 non-canonical NF-kB pathway	4.00 × 10^−5^	5
R-HSA-449147	Signaling by Interleukins	5.80 × 10^−5^	9
R-HSA-8877330	RUNX1 and FOXP3 control the development of regulatory T lymphocytes (Tregs)	6.27 × 10^−4^	2
R-HSA-8984722	Interleukin-35 Signaling	9.15 × 10^−4^	2

Abbreviations: HSA, *Homo sapiens*; TNF, Tumor necrosis factor; TNFR2, Tumor necrosis factor receptor; NF-kB, nuclear factor kappa-light-chain-enhancer of activated B cells; RUNX1, Runt-related transcription factor 1; FOXP3, Forkhead box P3.

**Table 3 ijms-25-09341-t003:** List of Treg-Associated Hub genes.

	Degree Centrality	Betweenness Centrality	Closeness Centrality	Clustering Coefficient
IL2RA	11	0.29	0.76	0.47
TRAF1	8	0.13	0.62	0.57
IL1R2	4	0.13	0.53	0.33
BATF	4	0.13	0.52	0.50
IL12RB2	4	0.13	0.50	0.53
CD80	9	0.12	0.64	0.69
TNFRSF4	9	0.06	0.64	0.69
TNFRSF18	9	0.06	0.64	0.69
TRAF3	6	0.03	0.50	0.73
CCR8	7	0.03	0.57	0.81

**Table 4 ijms-25-09341-t004:** Summary of the function of hub genes.

Classification	Gene Symbol	Function
Immune Regulation	IL2RA	IL2RA (Interleukin-2 Receptor Alpha) is crucial for the development and function of Treg cells [46]. It plays a role in the high-affinity IL-2 receptor, which is important for Treg cell proliferation and survival [47].
	TNFRSF4	TNFRSF4 (Tumor Necrosis Factor Receptor Superfamily, Member 4), also known as OX40, is involved in the activation, survival, and migration of Treg cells. It enhances Treg cell proliferation and function [48].
	TNFRSF18	TNFRSF18 (Tumor Necrosis Factor Receptor Superfamily, Member 18), also known as GITR, is important for Treg cell function and immune regulation. It contributes to Treg cell-mediated suppression and enhances their survival [49].
Cytokine Signaling	CD80	CD80 (Cluster of Differentiation 80) is involved in costimulatory signaling essential for T cell activation. It provides necessary second signals for T cell activation and survival through interaction with CD28 and CTLA-4 [50].
	TRAF1	TRAF1 (TNF Receptor-Associated Factor 1) plays roles in the downstream signaling of TNFRSF members, contributing to immune-response regulation and inflammation [51].
	TRAF3	TRAF3 (TNF Receptor-Associated Factor 3) is involved in various signaling pathways, including those triggered by TNF receptors, contributing to immune-response regulation and apoptosis [52].
	IL1R2	IL1R2 (Interleukin 1 Receptor Type 2) acts as a decoy receptor for IL-1, modulating immune and inflammatory responses by sequestering IL-1 and preventing it from interacting with the IL1 signaling receptor [53].
	IL12RB2	IL12RB2 (Interleukin 12 Receptor Beta 2) is involved in cytokine receptor activity and plays a role in the differentiation of Th1 cells, impacting immune cell signaling and response to infections [54].
Chemokine Signaling and Treg Migration	CCR8	CCR8 (C-C Chemokine Receptor Type 8) has a significant role in chemokine signaling, crucial for the migration and positioning of Treg cells within the tumor microenvironment [55]. Its high clustering coefficient (0.81) highlights its role in chemokine-mediated processes that guide Treg cells to their functional sites (Table 3).
Transcription Factors	BATF	BATF (Basic Leucine Zipper ATF-Like Transcription Factor) is involved in T cell differentiation [56]. It functions as a transcription factor that regulates the expression of genes essential for the development and function of various T cell subsets, including Tregs.

**Table 5 ijms-25-09341-t005:** Integrated Scoring for CCR8 and ICI Combination Therapy.

	Correlation with CCR8 Expression			Treg Infiltration Correlation	TMB	MSI	Clinical Status of ICIs	Final Score
	PDCD1	CD274	CTLA4					
BRCA	3	3	3	3	9	6	3	30
HNSC	2	2	2	3	6	6	3	24
COAD	2	2	2	2	6	6	3	23
STAD	2	2	2	3	6	5	3	23
THCA	2	0	2	3	4	3	0	14
READ	0	0	1	1	6	5	0	13

**Table 6 ijms-25-09341-t006:** Scoring Criteria for Correlation Analysis.

Criteria	Score	Significance Level	Description
Correlation (*p*-Value)			
*p*-value < 1 × 10^−50^	3	Highly significant correlation	Calculated the correlation coefficients and *p*-values for the relationships between CCR8 and PD-1, PD-L1, and CTLA-4
1 × 10^−50^ ≤ *p*-value < 1 × 10^−30^	2	Very significant correlation	
1 × 10^−30^ ≤ *p*-value < 1 × 10^−10^	1	Significant correlation	
*p*-value ≥ 1 × 10^−10^	0	Not significant	
TMB & MSI Correlation			
Correlation > 0.3	3	Strong positive correlation	Correlations between TMB, MSI, and hub gene to measure genetic instability and mutational burden within the cancer types
0.2 < Correlation ≤ 0.3	2	Moderate positive correlation	
0.1 < Correlation ≤ 0.2	1	Weak positive correlation	
Correlation ≤ 0.1	0	No correlation	

## Data Availability

Publicly available datasets were analyzed in this study. These data can be found here: GEO database (https://www.ncbi.nlm.nih.gov/geo/, retrieved on 24 July 2024).

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
