# Peer review of "A Bioinformatics Investigation of Hub Genes Involved in Treg Migration and Its Synergistic Effects, Using Immune Checkpoint Inhibitors for Immunotherapies"

_ijms, 2024, doi:10.3390/ijms25179341_

Round 1

Reviewer 1 Report

Comments and Suggestions for Authors

A very good bioinformatic analysis on Treg cells. While CCR8 has been already investigated to a certain extent for therapeutic purposes, it is good to summarize the influence of CCR8 expressing Tregs on immune responses. 

Out of scope of this paper, but it would also be interesting to investigate the found genes of interest in regard to being explicitly differentially regulated in Tregs or if some of the genes are in general differentially expressed in T cells as a whole class. 

As suggestion, it would be a good idea to maybe include how increased CCR8 on Tregs can correlate with PD-L1/PD-1 and CTLA4 regulation.

Furthermore, 2.5 and the beginning of 2.6 (first paragraph) are very repetitive in what you have done. Could be a good idea to revise this parts.

Altogether it is a very well written paper and very consise in data presentation.

Reviewer 2 Report

Comments and Suggestions for Authors

This study used GEO datasets of human and mice Tregs and looked for differential expressed genes between Tregs and nonTreg cells. Later, using prot-prot interaction analysis highlighted several hub genes. Among them CCR8, which was later correlated with immune-oncology markers of several cancers of the TCGA. The study is a pure bioinformatics exercise, there is no own data, and results are not validated. Therefore, it is somehow speculative results and conclusions. However, since results are based on previous knowledge (string dataset and immune-oncology markers) that we know are important, the final result of the study is positive.

 Additional comments (made in order as reading the manuscript):

 (1) Lines 31-39. You may mention that there are several subtypes of Tregs, the to main categories are natural-occuring Tregs and peripheral Tregs. Please refer to table 1 of this article that is quite useful.

 Front. Immunol., 05 January 2024. Sec. T Cell Biology. Volume 14 - 2023 | https://doi.org/10.3389/fimmu.2023.1291796

 (2) Lines 31-19. You may also describe that the most relevant gene is the FOXP3.

 (3) Line 43. You may describe that the role of Tregs may be different between carcinoma and hematological disease (in particular to lymphoid neoplasms).

 (4) Are Tregs CCR4 positive?

 (5) Line 45. What lymphoma subtypes are targeted with anti-CCR4?

 (6) Lines 46-53. Please cite the type of neoplasia of these studies.

 (7) Line 51-53. Could you please show the OS and PFS curves of these studies?

 (8) What is the ligand of CXCR4 and what cells express it?

 (9) Furthermore, selecting accurate targets for Tregs is difficult because markers such as FOXP3, CTLA-4, and PD-1 overlap with those of other immune cells. Do you mean that Tregs are positive for the 3 markers of FOXP3, CTLA-4, and PD-1? What non-Treg cells express FOXP3?

 (10) Line 81. Regarding the hub genes. “Hub” based on what type of statistical or bioinformatics analysis? (functional network association analysis, prot-prot interaction, etc.)

 (11) Line 89. Please describe briefly the 3 GEO datasets. Are they gene expression from Tregs?

 GSE128822: Transcriptional profiling of human tumor infiltrating Tregs (lung adenocarcinoma).

GSE116347: mice/human Tregs. Colorectal cancer Tregs.

GSE120280: Mice Tregs.

 (12)  Line 152. Why Treg migration?

 (13) In Table 1. What statistical test was used to calculate the p values and correlation coefficients?

 (14) In Table 1. Why these cutoff of p value and correlations? Why a P < 10exp-50 is highly significant? Should a correlation >0.8 be the strong one (I do not know what test or coefficient is used)?

 (15) The DEG are characteristic of Tregs (according to the authors’s analyses). Therefore, could you please explain what is the purpose of additional GO functional analysis? What is the logic of being related to “platelet activation” or “neutrophil migration”? Could you please explain if you looking for the function of Tregs?

 (14) Regarding section 2.4. The prot-prot string-based interaction is shown. This is fine, but, string integrates data from many sources and types of experiments. How specific are the results for Tregs?

 (15) Section 2.5 correlates CCR8 with several immune-oncology markers in several types of cancer from the TCGA. Could you please explain why CCR8 was chosen? Was CCR8 identified from the previous analysis of DEG of this study? Because it was identified in the hub (Table 5)?

 Barsheshet Y, Wildbaum G, Levy E, Vitenshtein A, Akinseye C, Griggs J, Lira SA, Karin N. CCR8+FOXp3+ Treg cells as master drivers of immune regulation. Proc Natl Acad Sci U S A. 2017 Jun 6;114(23):6086-6091. doi: 10.1073/pnas.1621280114. Epub 2017 May 22. PMID: 28533380; PMCID: PMC5468670.

 (16) Line 421. Regarding “Our study identified ten hub genes, including IL2RA, CCR8, TNFRSF4, TNFRSF18, and CD80, which play crucial roles in Treg function and migration”. If you are looking at DEG genes that are characteristic of Tregs, and used the string database, it looks logical relevant genes are being pointed out, doesn’t it? Did you find any new markers relevant for the Treg pathology?

 (17) Is it correct to mix human and murine data?        

 (18) Line 26. Regarding “This study highlights the potential of CCR8 as a biomarker and therapeutic target, contributing to the development of targeted cancer treatment strategies”. Comment: there is no doubt CCR8 is relevant marker for Tregs. However, what would be the effect of inhibition or activation of this markers using drugs in the pathogenesis and disease evolution in cancer? This study shows potentiality, however, it is a bioinformatics exercise, isn’t it?
